# Effects of mechanical interventions in the management of knee osteoarthritis: protocol for an OA Trial Bank systematic review and individual participant data meta-analysis

Erin M Macri ![ORCID],[1,2] Michael Callaghan,[3] Marienke van Middelkoop,[2] Miriam Hattle,[4] Sita M A Bierma-Zeinstra[1,2]

► Prepublication history and additional materials for this paper is available online. To view these files, please visit the journal online (http://dx.doi.org/10.1136/bmjopen-2020-043026).

[1]Department of Orthopaedics and Sports Medicine, Erasmus University Medical Center, Rotterdam, The Netherlands
[2]Department of General Practice, Erasmus University Medical Center, Rotterdam, The Netherlands
[3]Department of Health Professions, Manchester Metropolitan University, Manchester, UK
[4]School of Primary, Community and Social Care, Keele University, Keele, UK

**Correspondence to**
Dr Erin M Macri;
e.macri@erasmusmc.nl

## ABSTRACT

**Introduction** Knee osteoarthritis (OA) is a prevalent and disabling musculoskeletal condition. Biomechanical factors may play a key role in the aetiology of knee OA, therefore, a broad class of interventions involves the application or wear of devices designed to mechanically support knees with OA. These include gait aids, bracing, taping, orthotics and footwear. The literature regarding efficacy of mechanical interventions has been conflicting or inconclusive, and this may be because certain subgroups with knee OA respond better to mechanical interventions. Our primary aim is to identify subgroups with knee OA who respond favourably to mechanical interventions.

**Methods and analysis** We will conduct a systematic review to identify randomised clinical trials of any mechanical intervention for the treatment of knee OA. We will invite lead authors of eligible studies to share individual participant data (IPD). We will perform an IPD meta-analysis for each type of mechanical intervention to evaluate efficacy, with our main outcome being pain. Where IPD are not available, this will be achieved using aggregate data. We will then evaluate five potential treatment effect modifiers using a two-stage approach. If data permit, we will also evaluate whether biomechanics mediate the effects of mechanical interventions on pain in knee OA.

**Ethics and dissemination** No new data will be collected in this study. We will adhere to institutional, national and international regulations regarding the secure and confidential sharing of IPD, addressing ethics as indicated. We will disseminate findings via international conferences, open-source publication in peer-reviewed journals and summaries posted on websites serving the public and clinicians.

**PROSPERO registration number** CRD42020155466.

## INTRODUCTION

Knee osteoarthritis (OA) is a chronic musculoskeletal condition that affects approximately 24% of older adults.[1] It is associated with substantial pain, loss of function and reduced quality of life.[2] There are currently

---

## Strengths and limitations of this study

► We designed our protocol in collaboration with the osteoarthritis Trial Bank, an internationally recognised organisation with considerable individual participant data (IPD) experience, including established procedures for navigating the safe transfer and storage of IPD.

► IPD meta-analyses of randomised clinical trials enhance the ability to handle participant-level and study-level confounding, and increases the power to identify responder subgroups and mechanisms underlying treatment effects.

► A key limitation to undertaking IPD analyses relates to overcoming data-sharing hurdles, and the achievement of our aims will in part depend on the ability to successfully obtain IPD from eligible studies.

---

no known disease-modifying treatment approaches available for knee OA. Current guidelines recommend a core approach of exercise, education and dietary weight management if appropriate.[3] Adjunct interventions recommended for symptom management include pharmaceuticals such as non-steroidal anti-inflammatories, but also certain non-pharmacological interventions such as gait aids.[3] At the end stages of knee OA, when pain and disability become severe enough, knee arthroplasty is often undertaken. However, approximately 20% of individuals who undergo knee arthroplasty report not being satisfied following surgery.[4–6] With no effective disease-modifying treatment options, individuals often spend decades living with pain and disability.[7] It is thus clinically imperative to identify interventions that can contribute to symptom management in individuals living with knee OA.

Biomechanical factors, such as bony malalignment or poor movement patterns, may play a key role in the aetiology of knee OA by contributing to abnormal forces across affected joints.[8] Therefore, one broad class of interventions for knee OA involves application, or wear, of devices aimed at presumably improving an individual's biomechanics to reduce joint forces, improve symptoms and potentially modify the disease trajectory. Such interventions include gait aids such as canes, but also bracing, taping, orthotics and footwear, and they can easily serve as adjuncts to the current recommended core exercise-focused programmes.[3 9 10] This may be particularly relevant in individuals with certain biomechanical anomalies—such as frontal plane knee malalignment—because these individuals may be less likely to respond favourably to exercise.[11] These commonly prescribed treatments are relatively inexpensive, less invasive and have fewer side effects compared with other medical approaches such as intra-articular injections, oral medications or surgery.[12–14] Systematic reviews suggest some such mechanical interventions (eg, knee braces) may improve pain, though results have been conflicting or inconclusive regarding other outcomes or in other mechanical interventions.[12–21]

One possible reason for conflicting results is that knee OA represents a heterogeneous disease and certain subgroups with knee OA may respond better to mechanical interventions than others.[22–24] For example, lateral wedge insoles and insoles with subtalar strapping may be more effective in individuals with varus knee alignment,[13 14 17 21] while medial wedge insoles may be more effective in individuals with valgus knee alignment.[14] Including both of these groups of individuals in the same study could mask true treatment effects. While subgroup analyses within such a trial may successfully identify a 'responder' subgroup, most trials are not powered for such secondary analyses, and evaluating multiple possible subgroup characteristics further increases the likelihood of spurious findings.[25–27] Confirming the existence of such 'responder' subgroups could lead to identifying effective targeted biomechanical interventions in knee OA.

In addition to subgroup characteristics, another possible reason for conflicting results is the mechanism by which such interventions impart their effect. If the effect of mechanical interventions is mediated by a change in some biomechanical feature, then it may be that different types (eg, brace vs tape), design or dose of intervention will confer different outcomes via differing influence on the mediating feature.[28] Confirming the mediating role that biomechanics may play in mechanical interventions could help to optimise the design and application of mechanical interventions.

Several systematic reviews have evaluated the efficacy of mechanical interventions in knee OA, however, very little attention has been given to treatment effect modifiers or mediation analyses.[12–21] Moreover, to our knowledge, no study on this topic has yet to pool individual participant data (IPD) across studies. An IPD meta-analysis evaluates raw units of data rather than aggregate study-level data, and is thus a more robust approach to evaluating treatment effect modifiers and mediators.[29 30] Compared with traditional study-level meta-analyses, IPD meta-analyses of randomised controlled trials (RCTs) enhances the ability to handle participant-level and study-level confounding, and increases the power to identify responder subgroups and mechanisms underlying treatment effects.[30] The results using such an approach may, therefore, be more reliable and generalisable.[30]

Despite the growing recognition of the ethical and scientific importance of data sharing and scientific transparency, one of the biggest challenges in undertaking IPD analyses relates to overcoming data-sharing hurdles.[31–33] Barriers range from successfully reaching original study authors; willingness or ability of authors to share data; and international ethics and regulations issues.[31–33] The OA Trial Bank is an internationally recognised organisation that was established in 2010 and has developed procedures for navigating these barriers, including safely sharing, handling and storing IPD data from RCTs.[34] The OA Trial Bank steering committee supports and approves all projects, including providing input on research questions and study methods. This is, therefore, the ideal organisation in the field of clinical OA research to collaborate with in successfully conducting an IPD meta-analysis.

## AIMS

We aim to conduct a systematic review and IPD meta-analysis of RCTs, under the guidance of the OA Trial Bank, to evaluate the efficacy of mechanical interventions (ie, bracing, taping, orthoses, footwear or canes) in managing knee OA symptoms. Our primary aim is to identify subgroups of individuals with knee OA who respond favourably to mechanical interventions. Our secondary aim is to evaluate the effect of biomechanics as a mediator between mechanical interventions and symptoms.

## METHODS AND ANALYSES

The OA Trial Bank steering committee approved a summary of this study protocol prior to preparing the full study protocol,[34] and we published a basic study protocol with the International Prospective Register of Systematic Reviews.[35] The current detailed protocol has been written in accordance with the Preferred Reporting Items for Systematic Reviews and Meta-Analyses Protocols (PRISMA)[36 37] statement and PRISMA IPD (PRISMA-IPD) guidelines.[29] In collaboration with the OA Trial Bank, we will use methods described previously to guide the transfer and use of IPD,[38–40] updated recently to adhere to current European data-sharing regulations.[41]

We will search for relevant studies in five databases: MEDLINE, Embase, CINAHL, CENTRAL and Web of Science, with dates from inception to search date. An initial search was completed 21 August 2019, and we will update this search prior to beginning data analyses. We will develop a search strategy in collaboration with

an Erasmus MC librarian, using key words and medical subject headers, and adapting the syntax to the respective indexing vocabularies of each database (see online supplemental appendix 1).

We will identify studies that meet the following eligibility criteria:

## Participants

Adults (18 years or older) with knee OA (tibiofemoral or patellofemoral), diagnosed using any common method (eg, radiographs, MRI, clinical criteria, diagnosis by a healthcare professional). We will exclude studies where OA is determined by self-report alone. We will include post-traumatic OA, in particular knees with OA secondary to anterior cruciate ligament injury, regardless of whether or not they were previously reconstructed or repaired. We will, however, exclude knees with non-traumatic OA that have undergone major surgical procedures such as tibial osteotomy or total knee arthroplasty. We will exclude rheumatoid or other inflammatory arthropathies. If a study contains a subgroup of participants that meet our inclusion criteria, we will include that study if IPD data are retrieved, or if subgroup analyses are reported in the original publication.

## Interventions

Any intervention involving use or wear of mechanical devices (eg, bracing, taping, orthotics, footwear, cane) that is evaluated after more than 1 day or application of wear/use. We will include studies that combine these interventions with exercise or education/advice.

## Comparison

Any non-surgical treatment (eg, placebo, usual care, any other intervention that does not involve surgery), waiting list or no treatment.

## Outcomes

Our main outcome will be pain at the end of the study-specific primary duration of treatment. Treatment duration will be categorised as short term (<4 weeks), medium term (4–12 weeks) and long term (>3 months). Outcome measures will also be extracted at additional time points during treatment. We will not extract outcomes at any time points that are measured after discontinuation of treatment. Secondary outcomes will include function, quality of life, global perceived change, OA feature severity, biomechanics and adverse events.

## Study design

We will include peer-reviewed RCTs (or quasi-RCTs). We will exclude any other study design (eg, non-RCTs, prestudy and poststudy designs, observational studies). We will also exclude RCTs that only measure the acute effects of a single application of treatment.

## Languages

English, Dutch, German, French

Titles and abstracts will be initially imported into EndNote X9 (Clarivate Analytics) for deduplication, and then imported into Covidence for screening.[42] Two independent reviewers (EMM and MC) will screen titles and abstracts of all studies identified through this search strategy.[43] A third reviewer (MvM) will be consulted in the event of unresolved disagreements. Following the initial screening, two reviewers (EMM and MC) will independently review full text manuscripts to identify studies for inclusion in this review. A third reviewer (MvM) will again be consulted in the event of disagreements. We will review reference lists of included studies and relevant reviews for additional eligible studies.

For all included studies, we will contact the corresponding author by email. If a current email address cannot be found or the author does not respond (up to three attempts), we will attempt to reach them by other means (eg, phone, post, contact institution). Where IPD are available and authors or institutions are willing to share data, a data delivery agreement will be signed by both parties. Where local ethics regulations require it, ethics approval will be sought prior to sharing data. Pseudonymised or anonymised data sets (all formats are acceptable, eg, SPSS, Excel) and related data dictionaries will then be transferred and stored securely on a database at Erasmus MC, for use only as agreed on in the data delivery agreement. One original study investigator (first or senior author, at the discretion of the data owner) will be invited to be a coauthor of the project if they are willing to assume responsibilities that meet authorship guidelines.

We will convert all data sets to a common format, combine data sets with a new variable identifying original trial and harmonise variables. Data checking will include evaluating baseline characteristics and results of comparisons for our main outcomes against results reported in original publications. We will also check for balancing of baseline participant characteristics in each treatment arm, and evaluate the extent to which all randomised participants in the IPD datasets have been included in study analyses. Authors will be consulted in the case of any inconsistencies or discrepancies. In cases where discrepancies cannot be resolved, we will (on a case by case basis) either conduct a sensitivity analysis with that study removed, or we will exclude the study from our analysis altogether.

## Data extraction

Two independent investigators will extract data from all included published studies. From each study, we will extract the following data: country of study; funding source; study design; sample size; target population; inclusion/exclusion criteria; participant characteristics (age, sex, BMI, history of injury or surgery, comorbidities, psychosocial profile, metabolic profile, physical activity/fitness, lifestyle factors, medication use); type, dose and context of intervention (including compliance, cointerventions, protocol deviations, adverse events, drop-out

or withdrawal details); OA characteristics (compartment involvement, prevalence, severity, tissues involved), including pre–post if available; and pain, function, quality of life and biomechanics (eg, proprioception, knee alignment, strength, kinematics, kinetics) pre–post as available. Global perceived change will also be extracted as available. For all patient-reported outcomes, we will extract recall period in addition to the outcome. For all outcome measures, notably biomechanics, we will also extract whether scores and measures are taken with respect to the device applied/worn or removed. Where IPD are available, we will conduct all analyses using IPD instead of aggregate data, following data consistency checks described above.

## Risk of Bias

Two coauthors will independently evaluate risk of bias (ROB) for each included study using the Cochrane ROB tool V.2,[44] and disagreements will be resolved by a third investigator. The Cochrane ROB V.2 considers five domains of possible bias: randomisation; deviations from intended interventions; missing outcome data; measurement of the outcome and selection of the reported results. For each domain, ROB is rated as low, some concerns or high. Both the original publications and IPD datasets will be used for evaluating ROB. The overall study will be considered to be of low ROB if all five domains are rated as low ROB, and high overall ROB if at least one domain is rated as high ROB or if some concerns are identified in multiple domains. We will consult original authors in the event of inadequate reporting or inconsistencies.

## Statistical analyses

We will conduct an IPD meta-analysis of short term (<4 weeks), medium term (4–12 weeks) and long term (>3 months) effects of mechanical treatments (alone or in combination with exercise or education/advice) in comparison to other non-surgical treatments, sham, waitlist or usual care.

Where within-study missing data are sparse (less than 5%), we will assume data are missing completely at random and we will conduct complete case analyses, given the trivial loss of power and negligible implications on bias.[45] Where higher proportions of data are missing within a study, we will conduct within-study multiple imputation.[45 46] In cases where a variable was not collected in a given study, we will exclude that study from the relevant analyses.

## Treatment efficacy

To evaluate treatment efficacy, we will employ a two-step meta-analysis, first analysing each trial separately, and then pooling results across trials.[30 38 47] In step 1, within each trial, we will evaluate the effect of assigned intervention by intention to treat, regardless of method used in the original study. We anticipate that our main outcome, pain, will be evaluated differently across studies. To navigate this, we will evaluate the pain-related outcome from each study that ranks highest on the recommended

**Table 1** Hierarchy of pain-related outcomes proposed by Jüni et al[48]

| Rank | Pain outcome |
|------|--------------|
| Highest | Global pain score (eg, NRS, VAS) |
| | Pain on walking (same as one but task-specific) |
| | WOMAC pain subscale[59] |
| | Other composite pain scores (eg, KOOS Pain)[60] |
| | Pain on activities other than walking (eg, stair climbing) |
| | WOMAC global score (all three subscales combined) |
| | Lequesne osteoarthritis index global score[61] |
| | Other algo-functional composite scores |
| | Patient's global assessment |
| Lowest | Physician's global assessment |

KOOS, Knee injury and Osteoarthritis Outcome Score; NRS, Numeric Rating Score; VAS, Visual Analogue Scale; WOMAC, Western Ontario McMaster University Osteoarthritis Index.

hierarchy of pain-related outcomes to be used for meta-analyses[48] (see table 1). For each available time point (short term, medium term and long-term), we will fit an analysis of covariance (ANCOVA) model to obtain the treatment effect estimate, including baseline pain as a covariate.[49] We will report effect sizes from the ANCOVA model and their respective 95% CI. If study heterogeneity prevents us from harmonising pain data, then we will navigate this using a statistical approach based on available data. This will likely involve transforming data into standardised means differences or applying proportion of maximum scaling methods.[50]

In studies where we are unable to obtain IPD, we will extract aggregate data from published manuscripts as they are reported, for example, based on final scores or change scores.[51] Similar models will be performed for secondary outcomes as data permit. In cases of dichotomous outcomes, we will perform binary modelling and report effect sizes as relative risk (RR, 95% CI).

In step 2, we will perform random effects meta-analysis employing restricted maximum likelihood.[52] We will perform separate meta-analyses for each type of intervention (eg, braces, taping).[30 47 53] We will report study heterogeneity as $I^2$ and $\tau^2$.[54 55] In cases of notable heterogeneity ($I^2 > 50\%$),[55] we will consider possible sources such as device design, treatment duration, comparison treatment, treatment adherence or study quality. We will then consider performing meta-regression, subgroup analysis or sensitivity analyses to explain or account for these potential sources of heterogeneity. We will pool results of studies both with and without IPD data after verifying that effect sizes of IPD studies do not differ from non-IPD studies.[30 51]

In addition to a two-step meta-analysis, we will also perform a one-step meta-analysis as a sensitivity analysis.

In a one-step meta-analysis, all IPD datasets are harmonised into one large dataset and analysed together, with the addition of a covariate indicating original trial.

Within studies for which we have IPD and that report adherence to treatment, we will evaluate correlations between adherence and treatment effects. Where IPD are not available, we will extract aggregate data if reported. While we expect clinical and statistical heterogeneity to prevent meaningful meta-analysis of these data, we will pool data where possible.

### Treatment effect-modifier analyses

We will conduct treatment effect-modifier analyses to identify subgroups of individuals with knee OA who respond to various mechanical interventions.[38] We have proposed several subgroup characteristics that we hypothesise may modify the effect of mechanical interventions on our main outcome, based on expert opinion. These proposed subgroups include the following baseline characteristics: (1) mild vs severe OA (more severe joint space narrowing is associated with joint malalignment which may respond differently to mechanical interventions); (2) location of OA, specifically tibiofemoral versus patellofemoral OA, medial versus lateral tibiofemoral OA, or medial versus lateral patellofemoral OA (location of cartilage damage may be associated with differing joint alignment or source of symptoms); (3) varus versus valgus knee alignment (may be associated with different localised joint forces); (4) obese versus non-obese (may confer different amounts of mechanical stress) and (5) post-traumatic versus non-traumatic OA (possible different biomechanical anomalies). Where feasible, we will apply a two-stage approach, whereby we first investigate within-trial interactions within each study using IPD data, then pool results across trials.[27 53] This approach separates within-trial variation from across-trial variation, thus reducing the risk of ecological bias by analysing the effect of interest for individuals rather than groups of individuals.[27] We will conduct all treatment effect-modifier analyses in IPD data only.[53]

### Mediation analyses

We hypothesise that biomechanical factors may mediate the effect of these interventions (eg, kinematics, kinetics, proprioception, hypermobility) by reducing or normalising joint forces, which in turn reduces pain. If possible, we will conduct mediation analyses to evaluate this hypothesis.[38] We acknowledge that it is rare for studies to evaluate biomechanical variables both before-and-after treatment, so undertaking this analysis will depend on whether there are sufficient data available in included studies. If such an analysis is possible, we will employ a single mediator model to evaluate the proportion of the total effect of the intervention on pain that is mediated by a change in biomechanics.[56]

We will conduct funnel plot analyses where there are at least ten studies for a given intervention, to consider the possible effects of small sample size or publication bias.[57] We will summarise the overall level of evidence for each category of intervention using the GRADE approach.[58]

All analyses will be performed using Stata V.15.1 (StataCorp).

### Patient and public involvement

Patients and the public were not directly involved in the design of this study protocol. The OA Trial Bank advisory board includes patient members who provide overall input to the OA Trial Bank Steering Committee activities. We will solicit patient involvement through the OA Trial Bank advisory board and also through our institutional patient advisors (www.arthrosegezond.nl) for input in the analysis and interpretation of our study results, and to inform and guide dissemination of our study results.

## ETHICS AND DISSEMINATION

No new data will be collected, so de novo ethics approval is not required for our study. The OA Trial Bank has established protocols in place to guide the confidential and secure transfer and use of pseudonymised IPD[38–40] that adheres to current European data-sharing regulations.[41] We will collaborate with data deliverers to also adhere to relevant institutional, national or international regulations regarding data sharing and ethics. We will store all IPD datasets on a secure driver in accordance with OA Trial Bank procedures. We will disseminate findings via international conferences, open-source publication in peer-reviewed journals and summaries posted on websites serving the public and clinicians.

**Acknowledgements** We thank Dr. Wichor Bramer at Erasmus MC for assisting us with developing our search terms and managing our search.

**Contributors** The study was conceptualised by EMM, MvM and SMAB-Z. MvM and SMAB-Z provided specific guidance regarding OA Trial Bank procedures. Study design and methodology was performed by EMM, MC, MvM, MH, and SMAB-Z. Statistical expertise was provided by MH. EMM led the draft of the manuscript, and MC, MvM, MH, and SMAB-Z all provided substantial intellectually input to the manuscript. All authors approved the final manuscript.

**Funding** EMM was funded by a Canadian Institutes of Health Research (CIHR) Banting Postdoctoral Fellowship. The OA Trial Bank has received funding from ReumaNederland (grant number CIO-01).

**Competing interests** SMAB-Z reports grants from The Netherlands Organisation for Health Research and Development, CZ, European Union, Foreum, Dutch Arthritis Association, personal fees from Osteoarthritis Research Society International (OARSI), personal fees from Infirst Healthcare, and personal fees from Pfizer, all outside of the submitted work. We have no other disclosures.

**Patient consent for publication** Not required.

**Provenance and peer review** Not commissioned; externally peer reviewed.

**ORCID iD**
Erin M Macri http://orcid.org/0000-0003-2798-6052

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
