## [Reviewer comments · BMJ Open]

ARTICLE DETAILS

TITLE (PROVISIONAL)	The effects of mechanical interventions in the management of knee osteoarthritis: protocol for an OA Trial Bank systematic review and individual participant data meta-analysis
AUTHORS	Macri, Erin; Callaghan, Michael; van Middelkoop, Marienke; Hattle, Miriam; Bierma-Zeinstra, Sita

VERSION 1 – REVIEW

REVIEWER	Inga Krauß University Hospital Tuebingen, Dept. of Sportsmedicine, Germany
REVIEW RETURNED	09-Sep-2020

GENERAL COMMENTS	This protocol outlines the procedure of a systematic review and two-step meta-analysis on the efficacy of biomechanical devices for treating knee OA. The manuscript is well written, the procedure is comprehensible and results of the analysis will help to entangle contradictory results of previous studies and meta-analysis by use of IDP data and evaluations of effect modifiers that potentially outline treatment responders and non-responders. The collaboration with the OA Trial-Bank is a valuable prerequisite for the proper conduction of the planned meta-analysis with all its bureaucratic hurdles. Specific comments to the manuscript: Title; P3 L26-28; P6 L44 and elsewhere: You should mention that you also plan to use aggregated study data if no IPD are available. This is not true for modifiers and mediator but for the overall estimate of treatment efficacy. P3 L28: You mention 5 modifiers in the full text. Please change number accordingly. P8 L24-38: The search was already conducted (2019) – therefore the use of past tense would be more appropriate as changes to the search terms are no longer adjustable. P8 L39 et seq.: Will you also include subjects with self-reported OA (i.e. “has your doctor ever told you that you suffer from osteoarthritis at the knee?”)? Please state as in-/ or exclusion. P8 L39 et seq.: Are subjects included in case of one-sided joint arthroplasty and osteoarthritis on the other knee? Please specify. P8 L39 et seq.: Are subjects included after a corrective osteotomy at the tibia or femur? Please specify.
--

	P9 L19 et seq. - Outcomes: If I understand you right the evaluation will use outcome measures referring to the end of the treatment period (i.e. WOMAC 24h recall)? You mention that you are not interested in outcome measures after the treatment and you may therefore assume a direct influence of wearing the device on pain and other outcomes that diminishes after the offset of treatment? What I am getting at is that included data should be consistent in terms of a recall period of the instrument covering the last period of the intervention and not the period after it. Clarification would also be helpful regarding the biomechanical analysis. Are you interested in changes of biomechanics pre and post treatment (without wearing the device while measuring the biomechanical variables) or pre and end treatment (measuring the direct treatment effect of the device worn along a specific intervention period versus no device before treatment) or initiate and end treatment (if you are interested if the intervention period with the mechanical device changed biomechanics per se). This differentiation would also be helpful for the precise definition of the planned evaluation of a biomechanical effect moderator. P9 L19 et seq. – Outcomes: You report on compliance and adverse events in the data extraction sheet. From my perspective it would be beneficial to include safety (adverse events) as additional secondary outcome and treatment compliance (adherence/wear time or similar) as another effect moderator (responder-analysis). P10 L12: Figure 1 is missing. P11 L21: You may again mention studies combining these interventions with exercise or education/advice. i.e. "...mechanical treatments (as stand-alone or in combination with exercise or education/advice)..." P12 L19: Please specify sparse or higher proportion (i.e. percentage of missing data points). P12 L48: Please check if the following version is grammatically correct (I am not a native speaker...): "...and their respective 95% confidence intervals". P12 L50: Will aggregated study data will be weighted according to sample size? Please specify. P13 L40: You may also consider compliance as potential source of heterogeneity. P14 L35: As mentioned above you may also consider compliance (i.e. wear time) as a separate moderator for treatment effects. P14 L35 et seq.: Please specify as commented above (P9 L19 ff – Outcomes). P14 L53 et seq.: Does this two-stage procedure corresponds to the above mentioned two-steps procedure: 1st within-study analysis and 2° with all trials providing IDP? Could you please specify this a bit more for the common reader instead of referring to within and across trial variation only?
--	---

REVIEWER	Jennifer Hledik
REVIEW RETURNED	11-Nov-2020

GENERAL COMMENTS	This is a very well-written protocol. My comments for consideration are below. Aim (Page 6): You aim to evaluate efficacy of mechanical interventions in managing knee symptoms. How will you handle differences in what mechanical interventions aim to do? For example, medial wedge insoles versus lateral wedge insoles would target different knee OA populations and aim to have different biomechanical effects. Interventions (Page 8): How will you compare results of interventions that were possibly only used a very short time versus those used for a longer duration (as you will include any evaluated after more than 1 day of use)? Outcome (Page 8): How will you compare pain changes measured via different methods (e.g. WOMAC vs. VAS)? Statistical Analysis (Page 11): Will you correct for the comparison treatment? For example comparison to other non-surgical treatment may result in a smaller observed effect vs. comparison to usual care or a wait list. Treatment effect-modifier analyses (Page 13): What will be the criteria to determine mild vs. severe OA, if some studies do not have KL grading? Similarly, though PTOA vs. non-traumatic OA will be analyzed, what about primarily medial vs. lateral OA? One would not anticipate that the same mechanical interventions would be applicable to both (e.g. medial vs. lateral wedge). Mediation analyses (Page 13): How will you account for possible differences in targeted biomechanical effect (e.g. targeting frontal plane knee moment vs. alignment vs. reduced weight bearing, as with a cane).
---

VERSION 1 – AUTHOR RESPONSE

Reviewer: 1

Reviewer Name: Inga Krauß

Institution and Country: University Hospital Tuebingen, Dept. of Sportsmedicine, Germany

Comments to the Author

This protocol outlines the procedure of a systematic review and two-step meta-analysis on the efficacy of biomechanical devices for treating knee OA. The manuscript is well written, the procedure is comprehensible and results of the analysis will help to entangle contradictory results of previous studies and meta-analysis by use of IDP data and evaluations of effect modifiers that potentially outline treatment responders and non-responders. The collaboration with the OA Trial-Bank is a valuable prerequisite for the proper conduction of the planned meta-analysis with all its bureaucratic

hurdles. Response: Thank you very much for your thoughtful feedback. Below we address Reviewer suggestions point-by-point.

Specific comments to the manuscript:

Title; P3 L26-28; P6 L44 and elsewhere: You should mention that you also plan to use aggregated study data if no IPD are available. This is not true for modifiers and mediator but for the overall estimate of treatment efficacy.

Response: We have kept the title as is given that “Systematic Review” is a term that traditionally incorporates analyses of aggregate data, but have added this point elsewhere in the manuscript as suggested. Moreover, since the evaluation of efficacy is a secondary aim, we have chosen to clearly include reference to aggregate data in the methods pertaining to this aim.

Changes:

Page 2 line 13: *Where IPD are not available, this will be achieved using aggregate data.*

Page 6 line 19: We aim to conduct a systematic review *and* IPD meta-analysis of RCTs,

P3 L28: You mention 5 modifiers in the full text. Please change number accordingly.

Response: Thank you for catching this typo. We have now corrected ‘four’ to ‘five’ (page 2 line 14).

P8 L24-38: The search was already conducted (2019) – therefore the use of past tense would be more appropriate as changes to the search terms are no longer adjustable.

Response: Since the initial search was completed, we have not proceeded with any analyses because we would like the detailed protocol to have undergone peer review prior to doing so. Given the time that has passed, we plan to update our search prior to conducting analyses and publishing any findings. We have added a new sentence to reflect this.

Changes:

Page 7 line 10: We will search for relevant studies in five databases: MEDLINE, Embase, CINAHL, CENTRAL, and Web of Science, with dates from inception to search date. *An initial search was completed 21 August 2019, and we plan to update the search prior to beginning data analyses.*

P8 L39 et seq.: Will you also include subjects with self-reported OA (i.e. “has your doctor ever told you that you suffer from osteoarthritis at the knee?”)? Please state as in-/ or exclusion.

Response: Self-reported OA will not be included. We have now clarified this.

Changes:

Page 7 line 21: *We will exclude studies where OA is determined by self-report alone.*

P8 L39 et seq.: Are subjects included in case of one-sided joint arthroplasty and osteoarthritis on the other knee? Please specify.

Response: Thank you for pointing this out. We initially stated that we would exclude patients that had undergone arthroplasties, when in fact we should clarify that we will exclude *knees* that have undergone arthroplasties. We see no reason to exclude a participant with one arthroplasty if the contralateral knee that is studied meets eligibility criteria, because we are not conducting between-knee comparisons.

Changes:

Page 7 line 23: *We will, however, exclude knees with non-traumatic OA that have undergone major surgical procedures such as tibial osteotomy or total knee arthroplasty.*

P8 L39 et seq.: Are subjects included after a corrective osteotomy at the tibia or femur? Please specify.

Response: We will exclude knee OA where participants have a history of corrective osteotomy. We have now clarified this.

Changes:

Page 7 line 21: *We will include post -traumatic OA, in particular knees with OA secondary to anterior cruciate ligament injury, regardless of whether or not they were previously reconstructed or repaired. We will, however, exclude knees with non-traumatic OA that have undergone major surgical procedures such as tibial osteotomy or total knee arthroplasty.*

P9 L19 et seq. - Outcomes: If I understand you right the evaluation will use outcome measures referring to the end of the treatment period (i.e. WOMAC 24h recall)? You mention that you are not interested in outcome measures after the treatment and you may therefore assume a direct influence of wearing the device on pain and other outcomes that diminishes after the offset of treatment? What I am getting at is that included data should be consistent in terms of a recall period of the instrument covering the last period of the intervention and not the period after it.

Response: The Reviewer is correct that we will evaluate outcome measures that relate directly to application or wear of the device, and we will not extract data that relates to the period occurring after discontinuation of treatment. If we understand correctly, the comment here is specifically suggesting that the recall period of completed patient-reported outcomes should relate to the final period of treatment and not relate to any time following discontinuation of treatment. Thank you for this suggestion, we have now added that we will extract the recall period for all patient-reported outcomes in order to fully consider this.

Change:

Page 10 line 15: *For all patient-reported outcomes, we will extract recall period in addition to the outcome.*

Clarification would also be helpful regarding the biomechanical analysis. Are you interested in changes of biomechanics pre and post treatment (without wearing the device while measuring the biomechanical variables) or pre and end treatment (measuring the direct treatment effect of the device worn along a specific intervention period versus no device before treatment) or initiate and

end treatment (if you are interested if the intervention period with the mechanical device changed biomechanics per se). This differentiation would also be helpful for the precise definition of the planned evaluation of a biomechanical effect moderator.

Response: This is an excellent point and they are all interesting and important questions. As mentioned in the protocol (page 14 line 16) we recognize that it will be unlikely that many (if any) studies will measure biomechanics before and after treatment. However, we will extract all available data (i.e. under all conditions mentioned in the Reviewer's comment), including for each time point whether the device is applied/worn or not. If we have enough data to evaluate efficacy or conduct a mediation analysis, these conditions will have to be assessed separately.

With respect to evaluating whether biomechanics is an effect modifier, we only require baseline values of varus and valgus, meaning this would be prior to treatment and thus without application or wear of the device. This is described on page 13 line 23.

Changes:

Page 10 line 16: *For all outcome measures, notably biomechanics, we will also extract whether scores and measures are taken with respect to the device applied/worn or removed.*

P9 L19 et seq. – Outcomes: You report on compliance and adverse events in the data extraction sheet. From my perspective it would be beneficial to include safety (adverse events) as additional secondary outcome and treatment compliance (adherence/wear time or similar) as another effect moderator (responder-analysis).

Response: We agree that adverse events can be considered as a secondary outcome and have now added this to our list of secondary outcomes. We agree that adherence is an important consideration and we plan to extract adherence data. Adherence is not a baseline characteristic (though admittedly may relate to an underlying but unmeasurable character trait), so we don't believe that it fits our definition of effect modifier. However, we have now added some detail to the statistical analysis section on how we plan to consider adherence.

Changes:

Page 8 line 17: Secondary outcomes will include function, quality of life, global perceived change, OA feature severity, biomechanics, *and adverse events*.

Page 13 line 14: *Within studies for which we have IPD and that report adherence to treatment, we will evaluate correlations between adherence and treatment effects. Where IPD are not available, we will extract aggregate data if reported. While we expect clinical and statistical heterogeneity to prevent meaningful meta-analysis of these data, we will pool data where possible.*

P10 L12: Figure 1 is missing.

Response: Thank you for noticing this. Since we will be updating our search, we have decided for now to remove the preliminary PRISMA flowchart of search results from our protocol.

P11 L21: You may again mention studies combining these interventions with exercise or

education/advice. i.e. "...mechanical treatments (as stand-alone or in combination with exercise or education/advice)..."

Changes:

Page 11 line 10: ...effects of mechanical treatments (*alone or in combination with exercise or education/advice*) in comparison to...

P12 L19: Please specify sparse or higher proportion (i.e. percentage of missing data points).

Changes:

Page 11 line 13: Where within-study missing data are sparse (*less than 5%*),...

P12 L48: Please check if the following version is grammatically correct (I am not a native speaker...):

"...and their respective 95% confidence intervals".

Response: Thank you for catching this. We have made the suggested grammatical correction (page 12 line 4).

P12 L50: Will aggregated study data will be weighted according to sample size? Please specify.

Response: All meta-analyses will employ random effects, so sample size is considered in the weighting of each study.

P13 L40: You may also consider compliance as potential source of heterogeneity.

Response: We agree with this suggestion and have added it to our list of considerations (page 13 line 5).

P14 L35: As mentioned above you may also consider compliance (i.e. wear time) as a separate moderator for treatment effects.

Response: As mentioned above, for our effect modifier analyses, we are interested in evaluating baseline characteristics that might explain treatment subgroup effects. We have now described our handling of adherence measures in the statistical analysis section.

Page 13 line 14: *Within studies for which we have IPD and that report adherence to treatment, we will evaluate correlations between adherence and treatment effects. Where IPD are not available, we will extract aggregate data if reported. While we expect clinical and statistical heterogeneity to prevent meaningful meta-analysis of these data, we will pool data where possible.*

P14 L35 et seq.: Please specify as commented above (P9 L19 ff – Outcomes).

Response: We are not clear on what this comment means, but hope we addressed this sufficiently above.

P14 L53 et seq.: Does this two-stage procedure corresponds to the above mentioned two-steps

procedure: 1st within-study analysis and 2° with all trials providing IDP? Could you please specify this a bit more for the common reader instead of referring to within and across trial variation only?

Response: This is correct. This approach reduces confounding by trial and avoids what is called ecological bias – the Fisher 2017 reference walks through 3 different approaches to clearly illustrate the advantages of this approach.

Changes:

Page 14 line 6: Where feasible, we will apply a two -stage approach, *whereby we first investigate within-trial interactions within each study using IPD data, then pool results across trials (27, 55). This*

approach separates within-trial *variation from across-trial variation, thus reducing the risk of ecological bias by analysing the effect of interest for individuals rather than groups of individuals.* (27).

Reviewer: 2

Reviewer Name: Jennifer Hledik

Comments to the Author

This is a very well-written protocol. My comments for consideration are below.

Response: Thank you very much for your helpful feedback. Below we address each suggestion point-by-point.

Aim (Page 6): You aim to evaluate efficacy of mechanical interventions in managing knee

symptoms. How will you handle differences in what mechanical interventions aim to do? For example, medial wedge insoles versus lateral wedge insoles would target different knee OA populations and aim to have different biomechanical effects.

Response: This comment is insightful and we agree that meta-analysis will be challenging in cases where different types of interventions aim to achieve different mechanical effects. On page 12 line 17 we have reported that we aim to conduct separate meta-analyses based on 'type of intervention'. We have intentionally left this somewhat broadly defined because part of how we approach our analyses will depend on how many studies we find and which interventions are identified. In the case of the Reviewer's concrete example, it would likely make sense to conduct analyses that address two levels of inquiry: what is the effect of insoles that are designed to alter/correct frontal plane alignment (i.e. included together in one meta-analysis); and what are the effects of medial wedge insoles as distinct from lateral wedge insoles. This will be further complicated by whether those studies have recruited individuals with knee OA in general, or if they have targeted individuals based on baseline varus or valgus alignment and a hypothesized capacity to respond to treatment.

Interventions (Page 8): How will you compare results of interventions that were possibly only used a very short time versus those used for a longer duration (as you will include any evaluated after more than 1 day of use)?

Response: This is an important question and may help tease out whether there are optimal durations of treatment for different types of interventions. As above, how we handle this will in part depend on what types of studies we identify for inclusion. However, regarding time, we expect more than one level of analysis. First, we will conduct meta-analyses of study outcomes based on each study's primary end-point/treatment duration. Following this, we will look to see if meta-regression or subgroup analysis based on duration of treatment is indicated (as described beginning on page 13 line 4). In addition to primary endpoints, we are extracting data for all time-points collected within each study, and if possible and appropriate, we could also conduct meta-analyses at similar time points across multiple studies.

Outcome (Page 8): How will you compare pain changes measured via different methods (e.g. WOMAC vs. VAS)?

Response: We will only evaluate the highest ranking pain outcome in each study, as described in the

manuscript Table. Prior to conducting any analyses, we will harmonize all data. The exact steps to achieve harmonization of the pain data will depend on all the data sets that we receive and how pain has been measured and recorded. In our two-step analysis, the first step involves analyzing pain changes within each study using an ANCOVA (analysis of covariance) model. In the second step, a random effects meta-analysis will pool the treatment effects from the ANCOVA models in each trial. If study heterogeneity prevents us from harmonizing the pain data to our satisfaction, then our team, which includes an IPD statistician, will determine an approach based on available data. This will likely be one of two approaches:

1. If we have baseline balance in each trial, we can use a change score model and convert all study effect sizes into standardized mean differences (SMDs) prior to the second stage of meta-analysis.
2. If we do not have baseline balance in each trial, we can harmonise the pain outcomes using the proportion of maximum scaling (POMS) method, where:

$$\text{POMS} = \frac{\text{observed} - \text{minimum}}{\text{maximum} - \text{minimum}}$$
We will then analyse in the first step using an ANCOVA model and pool the treatment effects in the second step using a random-effects meta-analysis (Moeller J. *A word on standardization in longitudinal studies*:

don't. *Frontiers in Psychology*. 2015;6:1389. <https://doi.org/10.3389/fpsyg.2015.01389>).

Changes:

Page 12 line 4: *If study heterogeneity prevents us from harmonizing pain data, then we will navigate this using a statistical approach based on available data. This will likely involve transforming data into standardized means differences (SMDs) or applying proportion of maximum scaling (POMS) methods(50).*

Statistical Analysis (Page 11): Will you correct for the comparison treatment? For example comparison to other non-surgical treatment may result in a smaller observed effect vs. comparison to usual care or a wait list.

Response: In our primary analyses we do not plan to correct for comparison treatment. However, we agree that this is an important consideration, and we have listed type of comparison as one of the features we will consider evaluating in the event of high statistical heterogeneity (page 13 line 5). Pending these evaluations, it may be justified to conduct subgroup analyses or sensitivity analyses based on comparison treatments (particularly where the comparison treatment is another treatment such as braces vs. insoles, possibly resulting in smaller effect sizes as the Reviewer suggests here).

Treatment effect-modifier analyses (Page 13): What will be the criteria to determine mild vs. severe OA, if some studies do not have KL grading? Similarly, though PTOA vs. non-traumatic OA will be analyzed, what about primarily medial vs. lateral OA? One would not anticipate that the same mechanical interventions would be applicable to both (e.g. medial vs. lateral wedge).

Response: For each of the five potential treatment effect modifiers, we will only be able to include studies that report on the variable of interest. Thus, in studies where a measure of OA severity is not reported, we will not include that study in the effect modifier analyses. The comment about medial and lateral OA is well taken. We have therefore reworded this section to include the consideration of a higher fidelity of OA localization, should data enable this.

Changes:

Page 14 line 1: *...(ii) location of OA, specifically tibiofemoral vs. patellofemoral OA, medial vs. lateral tibiofemoral OA, or medial vs. lateral patellofemoral OA...*

Mediation analyses (Page 13): How will you account for possible differences in targeted biomechanical effect (e.g. targeting frontal plane knee moment vs. alignment vs. reduced weight bearing, as with a cane).

Response: This is a very good point and adds to the Reviewer's comment above regarding our study aim.

As mentioned in the manuscript (page 14 line 16), a mediation analysis might not be possible given that

RCTs of this nature which collect biomechanics data before and after treatment are uncommon.

Conducting mediation analyses (and meta-analyses) with biomechanics data could thus be limited. We aim to pool studies where similar biomechanical constructs were measured, otherwise we are limited to answering a very vague question as to whether mechanical devices (in general) confer their effect through changes in biomechanics (in general). It will be more likely that we will interpret the results of our overall study findings and generate hypotheses regarding mediating factors that we hope will drive future RCTs to include biomechanics outcome measures.

VERSION 2 – REVIEW

REVIEWER	Prof. Dr. Inga KRAUSS University Hospital Tuebingen Dept. of Sportsmedicine Germany
REVIEW RETURNED	14-Jan-2021
GENERAL COMMENTS	thank you for your point by point answers related to each of my remarks. I have no further comments as all of my previous aspects were considered in a comprehensible way. Best regards.
REVIEWER	Jennifer Hledik Stanford University, United States
REVIEW RETURNED	11-Jan-2021
GENERAL COMMENTS	The authors have addressed all of my prior comments - thank you!